# Evaluation of Climate Change Resilience of Urban Road Network Strategies

Siama Begum, Rachel S. Fisher *, Emma J. S. Ferranti and Andrew D. Quinn

Department of Civil Engineering, School of Engineering, University of Birmingham, Edgbaston, Birmingham B15 2TT, UK
* Correspondence: r.s.fisher@bham.ac.uk

**Abstract:** The impacts of the changing climate have caused extensive disruption to the road network in the United Kingdom in recent years. Roads are vital for economic growth and social wellbeing, and a disruption to the network can have disastrous consequences. Since the impacts of climate change will be felt at regional and local levels, it is the responsibility of local highway authorities to establish effective policies to strengthen the resilience of their section of the road network. This report uses the West Midlands as a case study and aims to evaluate its regional highway network management strategies, to determine the extent to which they promote resilience to climate change. Recommendations and findings from other literature are used to establish a set of evaluation criteria to compare the maturity of highway network management strategies for the West Midlands region. The evaluation of the policy documents is used to rank the maturity of the strategies, and recommendations are made to local authorities to highlight where the strategies could be improved. The analysis highlights the fragmentation and disparity between highways strategies across the region and consequently the vulnerability of the region to climate change.

**Keywords:** highways; transport policy; climate resilience; road networks; urban resilience; policy evaluation; multi-criteria analysis

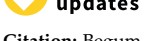



## 1. Introduction

Roads are the most used transport network in the United Kingdom (UK). The extensive road infrastructure network provides national and regional connectivity and enables multimodal journeys through access to railway stations, airports, and ports [1]. Two-thirds of commutes were made by car in 2018 [2]. In 2020 Heavy Goods Vehicles (HGVs) registered in Great Britain (GB) moved 136 billion tonne kilometres in the UK and the road freight sector contributed £13.6 billion to the UK economy in 2019 [3]. The role of road infrastructure is of strategic importance and disruption to the road network can have significant consequences for travelers, businesses and the wider economy.

Previous extreme weather events have highlighted the vulnerability of the UK road network [4,5]. For example, widespread and localised flooding over the winter of 2013/14 led to many road closures and major traffic disruption [6]. Moreover, disruption on the road network can have knock-on impacts for other sectors as infrastructure systems become increasingly interdependent [7,8]. As the frequency and severity of these extreme weather events increases due to climate change, it is essential that highway authorities devise strategies to ensure the resilience of the existing road infrastructure and adapt to the impacts of climate change [6].

The latest update from the International Panel on Climate Change (IPCC) indicates that the effects of climate change are already being experienced across all regions, worldwide, including hot extremes and intense precipitation [9]. This is evident across the UK as summer temperatures continue to set new records each year, with July 2022 recording unprecedented temperatures at over 40 °C [10]. These trends are likely to continue as it is

anticipated that global temperatures will continue to rise [11]. Additionally, it is predicted that sea levels may rise by one metre by the end of this century, future winters will be warmer and wetter, and extreme weather events will occur more frequently [12].

The 2017 Climate Change Risk Assessment (CCRA) has identified several risks that UK highways face as a result of a changing climate. Road infrastructure will be affected by increased thermal loading, changes in extreme wind events as well as landslips, and increased flooding [13]. Presently, approximately 6600 kilometres of UK roads are within regions prone to flooding, and this is anticipated to increase by up to 160% by the 2080s if adaptation measures are not implemented [14]. With an increase in flooding, there is also an increase in the risk of landslides triggered by severe rainfall events [13,15]. The cost of disruption due to flooding is high and totaled £200 million for flood events in 2007 [16]. Further risks from high temperature will lead to increased thermal loading on roads resulting in expansion, bleeding, and rutting, and additional maintenance will consequently be required to repair roads [17].

The National Infrastructure Commission (NIC) have undertaken a study on the resilience of the UK's infrastructure and have advised that infrastructure operators, such as National Highways (formerly Highways England), should develop long term strategies to improve the resilience of the road network [18]. Although National Highways have produced a Climate Adaptation Risk Assessment [19], and strategy [20], they are only responsible for the Strategic Road Network (SRN) which includes all motorways and some 'A' roads. The SRN accounts for only two percent of all roads in England, however motorways and 'A' roads convey over 60% of road traffic [1].

Most of England's roads are managed by local highway authorities, who have a responsibility to maintain their section of the network [20]. Despite this the resources and capacity that are required to deliver climate change adaptation at a regional level are not being assessed as there is a tendency to focus on national networks within CCRAs. However, it is expected that future CCRAs will better account for risks and stakeholders at a regional level by involving stakeholders such as local highway authorities, who play a significant role in achieving local transport resilience [21].

The Department for Transport has advised that local authorities should prepare for, and be able to respond to, extreme weather events [6]. Some local authorities, such as Cheshire East, have responded to these recommendations by identifying a Resilient Highway Network that consists of critical routes, which will be prioritised in terms of investment and maintenance [22]. However, many local authorities are not conducting assessments to measure the efficacy of their resilience strategies, and in several cases, clear resilience strategies have not been defined.

Since nearly every journey, regardless of the main transport mode, begins or ends on a local road [23], it is vital to evaluate the strategies proposed by local highway authorities to determine how prepared the road network infrastructure is for future climate change. To this end, this paper presents a novel methodological approach to evaluate the inclusion of climate change resilience criteria within strategic planning and policy documentation. The method developed is scalable, tractable, and agile as it can easily be applied to policy beyond transport and at different organisational and geographic scales. This paper presents the results of the application of the developed methodology to regional highway network management strategies.

This study aims to evaluate the maturity of regional highways infrastructure policy within the context of climate change resilience. The objectives are to:

1.  Conduct a review of academic literature to inform evaluation criteria.
2.  Systematically identify highway network management strategies for the West Midlands region.
3.  Conduct a multi-criteria evaluation of regional road network management policies and rank their maturity and resilience.
4.  Identify recommendations to local highway authorities of best practice.

Following this introduction (Section 1), this paper establishes the current understanding of the term resilience in relation to weather and climate impacts on transport, an overview of the role of transport policy, particularly at a regional level, as well as how resilience can be assessed (Section 2). Section 3 establishes the evaluation criteria used to score and rank the key aspects of resilient infrastructure policy and how the regional highways policy documents were identified. Section 4 presents the results of the evaluation of 33 regional policy documents from the case study area, the West Midlands region in the UK. The outputs of the analysis are then discussed within the context of a changing climate (Section 5), followed by reflection on the approach developed, its limitations and future work. Finally, concluding remarks are made in Section 6.

## 2. Literature Review

Initially used in the study of ecology, the term 'resilience' has gained relevance in several fields over the last forty years, including psychology and engineering [24]. In the context of transportation, resilience is defined as the ability of a transport network to absorb and recover from disturbance, such as the impacts of an extreme weather event, whilst retaining its function and continuing to operate [22,25]. There have been many approaches taken to describe the different elements of resilience [18,26,27], however the Cabinet Office [28] approach is often adopted by UK transport stakeholders, considering four components of resilience, as illustrated in Figure 1 [29].

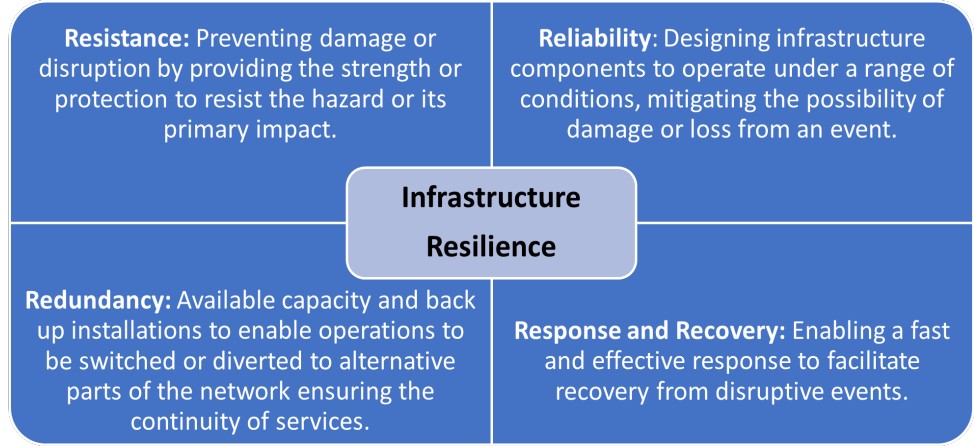

**Figure 1.** The four components of infrastructure resilience (adapted from [28] and reproduced from [29] with the author's permission).

Resistance is concerned with strengthening or protection of the transport network to withstand the impacts of weather hazards. These approaches are often flawed as this protection is provided based on historic events rather than by preparing for conditions that can be expected in the future and are more severe than those previously experienced. Resistance to climate change is particularly challenging because extreme weather events are not always anticipated, and therefore protection measures may be insufficient [28].

Reliability refers to the ability of the transport network to provide the required service level when exposed to a broad range of conditions. Unfortunately, specification of environmental conditions is often lacking as the effect of climate change on these ranges is often not considered [30], hence the reliability of the system may not be guaranteed.

Redundancy describes the capacity of the network and the extent to which there are alternate means available to enable services to continue to run in the event of a disruption. For example, in the event of a road closure due to flooding, a road network with redundancy would have sufficient capacity to enable road users to take an alternative route to complete their journey [31].

Finally, response and recovery is a component of resilience that focuses on the ability to respond to a disruption and restore planned service levels in a timely manner. This

often requires thorough planning and preparation across multiple government bodies (e.g., emergency responders) before the event occurs [28,32].

As the climate will continue to change for the foreseeable international, national and regional transport systems will not be resilient to every weather event and natural disaster that occurs over the coming years and decades. Despite this, all four characteristics of resilience should be incorporated into transport networks as they are critical to the socio-economic success of individuals, communities, and businesses [6].

The importance of climate resilient transport infrastructure is recognised within the United Nations' 17 Sustainable Development Goals (SDGs) and most notably in SDGs 9, 11 and 13. SDG 9 refers to the building of resilient infrastructure, such as transport systems as well as sustainable industrialisation. SDG 11 refers to resilient cities of which transport networks are a key component. Action to address the impacts of climate change are referenced in SDG 13 which therefore includes all systems including transportation [33]. These SDGs are increasingly influencing policy such as the European Union's Sustainable Urban Mobility Plans (SUMPs) which aim to ensure high quality and sustainable mobility across whole functional urban areas [34]. The guiding principles behind SUMPs have developed since their conceptualisation, however they do not currently explicitly incorporate climate resilient infrastructure. To work towards increasing the resilience of their services to climate change, it is important that transport operators and local authorities take a sustainable and proactive approach to identify and address vulnerabilities and develop resilient transport policies [18,33].

### 2.1. Policy for Transport Resilience

The purpose of transport policy is to establish a set of proposed actions to improve the performance of the transport network and other social, environmental, and economic factors [35]. Transport policy aims to articulate how resources will be allocated, identifies systems that should be prioritised for investment and coordinates actions across sectors, whilst protecting the interests of its various stakeholders [36]. Policymaking can be a complex task in the transport sector due to the challenges of a growing population, the urgency to meet SDGs and rapid advancements in technology that must be considered. In addition, transport networks are subject to a range of complex shock events and disturbances, and due to the increasing interdependencies between systems, these risks are compounded [13]. It is therefore vital that policies are adapted to meet the needs of the future, due to the fundamental role that transportation plays in economic growth and social wellbeing. Furthermore, as the frequency of extreme weather events increases because of climate change, policies must be updated to focus on strengthening the resilience of transport networks [37–39].

Where policy is in place, its feasibility and efficacy should be evaluated. This is challenging as the evaluation criteria used is dependent upon what is deemed important in each community, sector, or other circumstance [37]. In the case of infrastructure policy, some criteria that are commonly used for comparison are shown in Table 1. When comparing and evaluating other types of policy at different levels within society, do these remain relevant and how might resilience be prioritised?

**Table 1.** Criteria commonly used to compare infrastructure policy instruments (Source: adapted from [37]).

| Criterion | Explanation |
| --- | --- |
| Workability | Considering technical and political feasibility, administrative requirements, resources required for implementation and enforceability. |
| Effectiveness | The ability of an instrument to effect the desired change based on the behavioural assumptions of target population. |
| Efficiency | This is often calculated based on cost–benefit analysis or cost-effective analysis. |

**Table 1.** *Cont.*

| Criterion | Explanation |
|---|---|
| Equity | Based on the distributive effects of policy, both in time and in space, such as the allocation of benefits and risks, the possibility of entrenching or reducing disadvantage and who bears the cost of externalities. |
| Appropriateness | Considering proportionality, as well as the broader cultural context of the policy. |
| Timeliness | How quickly the tool works relative to the urgency of the problem. |
| Flexibility | Whether the tool is sufficiently flexible to adjust to changing circumstances. |
| Accountability | The extent to which those implementing the instrument can be held accountable for their actions. |
| Choice | The degree to which citizens have choice in the policy. |

### 2.2. Regional Transport Resilience

Climate change is a global challenge and policies regarding the subject are being discussed at a national and international level [40,41] as exemplified by the UK (Section 1). However, most impacts of climate change will be felt at regional and local levels, therefore it is crucial to enhance the resilience of regional road networks [42]. Local authorities are often the main implementing bodies for many climate change policies to achieve goals that are set by national government, indicating the significant role of local authorities in the design and implementation of climate policies [43].

The West Midlands is the largest urban area in the UK, excluding London [44], despite this, there are currently no strategies that focus specifically on the climate resilience of its road network. The transport network in the region is managed by Transport for West Midlands (TfWM) which is part of the West Midlands Combined Authority (WMCA). There are seven local authorities within the TfWM: Birmingham City Council (CC), City of Wolverhampton Council (CWC), Coventry CC, Dudley Metropolitan Borough Council (MBC), Sandwell MBC, Solihull MBC, and Walsall Council. The TfWM partnership are responsible for over 9000 kilometres of roads [45] and the current cost of road congestion to Birmingham's economy is £632 million per year [46]. In 2017, 8 million journeys were made every day within the region [47] and it is predicted that congestion levels will increase by as much as 83% by 2035 [48]. Disruption to transport in the West Midlands from one flood event has been predicted to cost between £30 million and £80 million [49]. These figures are from a 2004 study, the most recent available, highlighting the lack of action to understand the impact of climate change at a local level.

### 2.3. Assessing Transport Resilience

In order to determine whether climate change resilience policies are effective, they must be assessed. Academics have explored ways to assess the resilience of transport networks, both quantitively and qualitatively. In addition, other work has focused on the effectiveness of policy to support the resilience of transport. This section reviews the challenges of applying previously developed approaches to the assessment of regional road networks and infrastructure policy.

Gordon and Matheson (2008) proposed measures that could be used to assess the resilience of road networks, such as the resistance of the network to disruption [50]. This could be assessed by examining the extent of damage and level of functionality of the network after the event has occurred. It may also be assessed by considering factors such as the presence of other vulnerable services in the road corridor, the failure of which could cause disruption to the road network. These interdependencies with other sectors, such as energy and water, should be addressed in resilience strategies because they can exacerbate failures [18,51]. Another parameter identified is the layout of the network and whether there are alternate routes available to allow the network to continue to function despite a shock or stress. Other measures include the time that is required to restore the

network and the volume of traffic, since the disruptions will have greater consequences for parts of the network with higher traffic volumes. These parameters can be useful to inform the evaluation of resilience strategies because if a network has redundancy and has alternative routes available, road users are still able to make their journeys despite a disruption. However, there should be a clear plan in place to inform them of extreme weather conditions and allow them to make their own travel decisions. They may choose to use an alternate mode of transport, take a different route, or not make their journey at all [50].

One study has established a Transport Resilience Indicator Framework [31], which used qualitative measures and identifies six dimensions of transport resilience assessment: engineering, services, ecological, social, economical, and institutional. The 'engineering' dimension is associated with the robustness, replacement, and rapidity of restoration of the network. Robustness refers to the ability of the network to withstand the impacts of extreme weather events and the study indicates that it is concerned with the design of the road infrastructure in the study area in New Zealand. In the UK, the review of design standards is not directly relevant to the evaluation of regional strategies because local highway authorities use national design standards for the design of local roads. These standards are overseen by National Highways, who are responsible for updating the standards to prepare for the expected effects of climate change [19]. However, since local authorities are responsible for road maintenance as well as the replacement or upgrade of vulnerable sections of the local road networks, regional resilience strategies should clearly outline roles and responsibilities between the national and regional highways stakeholders. This will ensure that the level of resilience that is expected to be delivered is understood as potential events are anticipated [18].

The 'services' dimension of the Transport Resilience Indicator Framework relates to the redundancy of the network which is a useful measure to inform the evaluation of resilience strategies as it accounts for travel behaviours which can be significantly influenced by weather [52]. The role of communication channels have been identified as a useful tool to manage travel behaviour during adverse weather [6], this includes motorway signals, information boards, televised media, and digital platforms such as social media. However, in the UK this is often undertaken at a national and regional level rather than at a local level. Furthermore, the engagement of local stakeholders, communication with and knowledge exchange between these parties can be beneficial to understanding and mitigating risks associated with weather hazards and climate change impacts on highways [53,54].

Additionally, the 'institutional' dimension of the Transport Resilience Indicator Framework is associated with collaboration, learning, policy, and the allocation of resources. Collaboration can be effective as local highway authorities can work together, share information and best practices, and identify critical routes and vulnerabilities to enhance the resilience of the road network [31]. The study also highlights that it is important to review and adapt contingency plans, particularly after a disruption has occurred, to reflect on lessons learned. This is applicable to the evaluation of regional strategies as they should be reviewed to ensure they are not outdated, although it should be emphasised that they should not just be reviewed following a disruption but also periodically to prepare for uncertain or unexpected events [38]. Furthermore, in terms of the 'economic' indicators, the cost of enhancing the resilience of the road network should be accounted for when funding is allocated [31]. This is relevant when evaluating strategies because the strategies should be affordable and informed by the full cost of response [18,55].

Various approaches have been developed to measure and assess the resilience of transport networks, including quantitative methods. Serulle (2010) defines a method that is based on a 'resiliency cycle' that comprises of four stages: normality, breakdown, annealing, and recovery [56]. Breakdown is a measure of the damage caused to the network due to a disruption and annealing and recovery refer to how quickly the network can be restored back to normality. A disruption to the transport network is then investigated using a set of variables and corresponding metrics, including the average delay encountered and

the capacity of the network, to measure its resilience. This method focuses on returning to normality but does not consider how to take a more proactive approach to improve resilience once services are restored. It does not recognise the importance of learning from past disruptions or the role that government bodies and local authorities play to enhance resilience. Therefore, the measures identified in this study are not useful for the evaluation of resilience strategies because they focus on achieving short-term resilience rather than acknowledging that achieving resilience will require a long-term collaborative effort [56].

Hughes and Healy (2014) developed a framework to measure the resilience of transport networks that suggests a range of measurement categories based on six resilience principles [57]. The following principles are identified: robustness, redundancy, safe-to-fail, change-readiness, networks, and leadership and culture. The 'change-readiness' category is particularly noteworthy because it encompasses a range of resilience measures, such as effective communication to warn road users of disruptions, and the use of technology to monitor events and communicate data. In addition, it highlights that clear roles and responsibilities should be defined, funding should be allocated for all elements of resilience planning, and past actions should be reviewed to determine their success at combatting hazards, all of which are useful indicators to evaluate resilience strategies. Additionally, the framework recognises the significance of building effective partnerships so that local authorities can work collaboratively towards a shared goal of enhancing resilience [51,57].

A study conducted by Markolf et al. (2019) suggests that flexibility should be considered in the assessment of transport resilience due to the evolving and complex threats that transport networks face [30]. Flexibility is defined as the ability of a network to accommodate and adapt to foreseeable changes and uncertainty [58] and there are emerging examples of how this is being incorporated in the transport sector. For instance, it is proposed that if autonomous vehicles are introduced in the future as public transport, they will not follow defined routes which will provide them with modal flexibility as they will be able to avoid disruptions. This is a useful parameter when evaluating transport resilience strategies as they should be sufficiently flexible to adjust to changing circumstances [37]. Furthermore, the research paper highlights the importance of considering the growing interdependencies between the transport and power sectors, especially as the UK has targets for every car and van to be zero emission by 2050 [59], implying that electric vehicles will become increasingly prevalent. The risks associated with climate change as well as power outages will introduce additional vulnerabilities to road networks, hence the interdependencies between sectors must be addressed when evaluating resilience strategies.

Existing policies can quickly become outdated and should be adapted to address current challenges that may be more diverse and complex than previous ones [36,38]. Additionally, it should be possible to update and improve policy particularly as the full extent of the change in the earth's climate is still unclear and the scale of seasonal variation at a regional level is still uncertain. Policies should be reviewed regularly to determine whether any modifications are necessary or whether they are still required, and they should define clear roles and responsibilities [41]. Furthermore, transport networks are becoming increasingly interdependent with other infrastructure systems which can exacerbate failures, hence these interdependencies should be addressed in effective and resilient policies [18].

## 3. Methodology

Following the review of academic literature, eight criteria for effective transport, and particularly highways, policy have been identified as discussed in the previous section. A critical analysis of the aspects of effective transport policy with regard to delivering resilient road networks and highways infrastructure highlighted key criteria. A number of these commonly used criteria have specifically been identified as important in the pursuit of resilient infrastructure and consequently should be embedded into infrastructure policy [18,37]. A summary of these criteria is presented in Table 2.

**Table 2.** Evaluation criteria for comparing resilient infrastructure policies.

| Evaluation Criteria | Interpretation | References |
|---|---|---|
| Roles and responsibilities | The position of each stakeholder and tasks are clearly attributed to them so that they can be held accountable. | [18,41,51,57] |
| Regularly reviewed and updated | The revision schedule is aligned with appropriate timescales (e.g., new climate datasets). | [18,31,38,41] |
| Cost–benefit appraisal | The implementation of adaptation actions and processes is evaluated with regard to both costs and benefits. | [18,31,37,55] |
| Accessibility | Describes the ease with which the document can be viewed and understood by a range of actors. | [18,37,53,54] |
| Communication with road users | Identified channels through which local authorities and other regional transport stakeholders can communicate weather conditions and other hazards with road users. | [22,53,54,57] |
| Collaboration | Working together, sharing information and best practices for climate resilience. | [31,38,56,57] |
| Flexibility | Whether the policy is sufficiently flexible to adjust to changing circumstances. | [6,30,37,58] |
| Interdependencies | Awareness of other urban systems that are dependent upon the roads as well as those systems upon which roads are dependent. | [13,18,50,51] |

Building on other approaches these evaluation criteria for comparing resilient infrastructure policies will form the basis of a multi-criteria analysis which will be used to evaluate the regional road network management policies [24,60]. The highways policy documents will be scored on a scale of 1 to 5 (with 1 being the worst and 5 being the best) for each of the criteria as shown in Table 3. The total across the 8 categories will be the policy's overall score. As this multi-criteria analysis only employs 8 criteria it was not deemed to be complex enough to warrant the need for a weighting to be applied to the analysis. It is beneficial to introduce a weighting in more complex scenarios, but here it is not essential [61]. However, future work may benefit from a sensitivity analysis to determine criteria weightings if the range of criteria is extended.

Building on existing approaches [24,60], a systematic selection process is performed to identify regional road network management policy documents. This procedure consists of four stages (1) systematic selection using an online database, (2) the results are filtered using specific criteria, (3) further screening by title, keywords, and abstract, and (4) in depth analysis of the policy document. As there is no dedicated online database for local government policy documentation, the systematic search for local policy documentation was undertaken using Google Search. Local authorities commonly have websites where they present information relating to their management of local highways. Google Search is used to find the following criteria and keywords: "road" OR "highway", AND "strategy" OR "plan" OR "framework" OR "assessment" OR "policy" OR "guidance", AND "Birmingham City Council" OR "Coventry Council" OR "Dudley Council" OR "Sandwell Council" OR "Solihull Council" OR "Walsall Council" OR "Wolverhampton Council" OR "West Midlands". The term "climate resilien *" is not included in the search because local authorities in the West Midlands do not have dedicated climate change resilience strategies for their road networks. At the end of this stage, 3,690,000 search results are generated.

**Table 3.** Scoring criteria to evaluate regional policy documents.

| Evaluation Criteria | Worst<br>1 | 2 | Score<br>3 | 4 | Best<br>5 |
|---|---|---|---|---|---|
| Roles and responsibilities | Roles and responsibilities are not identified | The roles and responsibilities of the local authority are briefly indicated. | Some of the roles and responsibilities have been identified, but not clearly or extensively. | The roles and responsibilities of the local authority are clearly defined, including the department. | Roles and responsibilities for internal and external stakeholders are clearly defined. |
| Regularly reviewed and updated | No review process or intention to update documentation. | A monitoring process is indicated without reviews and updates. | States that it will be reviewed but does not specify that it will be updated. | States that it will be regularly reviewed and updated. | Identifies how often it will be reviewed and updated. |
| Cost–benefit appraisal | No evidence of the costs or benefits being considered. | Only brief consideration of funding, but no sources are listed. | Various funding sources are identified for the delivery of schemes. | Consideration of the scheme costs in relation to the resulting benefits. | Evidence that a full cost–benefit analysis has been conducted. |
| Accessibility | Not accessible and difficult to use and understand. | Not easily accessible without prior knowledge. | Accessible on the local authorities' website but may not be understood by all stakeholders. | Accessible on the local authorities' website. The strategy can be understood by key stakeholders. | Easily accessible and easy to use and understand by a range of stakeholders. |
| Communication with road users | No consideration of communicating weather conditions to road users. | Acknowledgment of the importance of sharing information with the public. | Acknowledgment of the importance of sharing information of adverse conditions with the public. | Indication of communication actions, but with no reference to methods. | Methods to effectively communicate weather conditions to road users identified. |
| Collaboration | No aspect of collaboration with other authorities. | The strategy briefly recognises that collaboration is required, but the collaborators are not identified. | The strategy recognises that collaboration is required and identifies some of its collaborators. | Evidence of collaboration with other authorities and organisations. Identifies some of its collaborators. | There is strong evidence of collaboration with other authorities and organisations. |
| Flexibility | No elements of flexibility incorporated. | Briefly incorporates elements of flexibility. | Indication of modal flexibility and changing travel behaviours. | Modal flexibility and changing travel behaviours is encouraged to increase capacity in the short-term. | Incorporates elements of flexibility to enable the road network to accommodate and adapt to long-term changes. |
| Interdependencies | Interdependencies with other sectors have not been identified. | Acknowledges interdependencies but not with which sectors. | High level interdependencies outlined with two sectors. | Interdependencies with more than two sectors described in detail. | Comprehensive review of interdependencies with other sectors. |

In the next stage, the results are refined by date to exclude documents that were published before 2010, by language to find documents written in English, and by region to identify documents that have been published in the UK, as this is the country of interest for this study. At this point, 399 results are obtained. During the penultimate stage, the titles are checked to exclude documents that refer to national policies or local authorities outside of the West Midlands. Finally, the remaining documents are reviewed in depth to only include road network management strategies for the seven specified local authorities. This narrows down the results to 33 policy documents and reference key, which can be seen in the Supplementary Materials (Reference List S1 and Table S1 respectively).

## 4. Results

The criteria outlined in Section 3 are used to evaluate how effectively a range of regional highways policy documents capture the aspects of resilience identified in the literature review. The full results of the policy document evaluation can be seen in the Supplementary Materials (Table S2) along with the policy document key (Table S1). The performance of the policies against each of the criteria can be seen in Table 4 which shows the average score across all the policies in the West Midlands region. Collaboration is shown to be the most highly scoring area, whilst interdependencies scored the lowest. The average total score across the categories is 25.2 (maximum of 5 per criteria evaluated, and 40 total).

**Table 4.** Overview of results of the evaluation of policy documents by assessment criteria.

| Resilience Characteristic | Policy Document Average Scores |
|---|---|
| Roles and responsibilities | 3.3 |
| Regularly reviewed and updated | 3.9 |
| Cost–benefit appraisal | 2.8 |
| Accessibility | 4.5 |
| Communication with road users | 2.6 |
| Collaboration | 4.7 |
| Flexibility | 3.2 |
| Interdependencies | 1.4 |
| Average Total Score | 25.2 |

Table 5 shows these results by local authority or organisation, the WMCA and TfWM organisations are in the same category (since TfWM is the transport arm of the WMCA) and the Black Country refers to collaboration and joint policy shared by the CWC, Dudley MBC, Sandwell MBC, and Walsall Council. Table 5 shows the worst performance at 21.4 by Coventry CC and the highest performance from the Black Country with an average policy document score of 30. The individual policy document resilience evaluation scores can be seen in Figure 2, which shows the distribution of the scores across the local authorities. Strategies produced through the collaborative Black Country and WMCA/TfWM performed well, whereas Coventry CC and Dudley MBC were amongst the lower performing authorities, as shown in Table 5. The Black Country scores the highest for an individual policy with 30 points for their Local Strategy for Flood Risk Management [62]. While the Birmingham CC and Coventry CC Winter Service Plan are joint lowest scoring with only one point each due to them not being accessible.

The average policy score is indicated in Figure 2 at 25.2 points, which policy documents W-AI fell short of (13 documents out of a total of 35). Four of the 13 policy documents performing below average belonged to Walsall Council, which is more than half of their documents (Walsall Council had a total of 7 policy documents evaluated). Dudley MBC's policies both performed below average whilst the WMCA/TfWM were the only organisations whose policies consistently performed above average.

**Table 5.** Overview of results of the evaluation of policy documents by local authority or organisation.

| Local Authority/Organisation | Average Policy Score |
|---|---|
| Birmingham CC (or BCC) | 24.5 |
| Black Country | 30.0 |
| Coventry CC (or CCC) | 21.4 |
| Dudley MBC | 23.5 |
| Sandwell MBC | 26.0 |
| Solihull MBC | 27.4 |
| WMCA/TfWM | 27.8 |
| Walsall Council | 24.4 |
| Average Total Score | 25.2 |

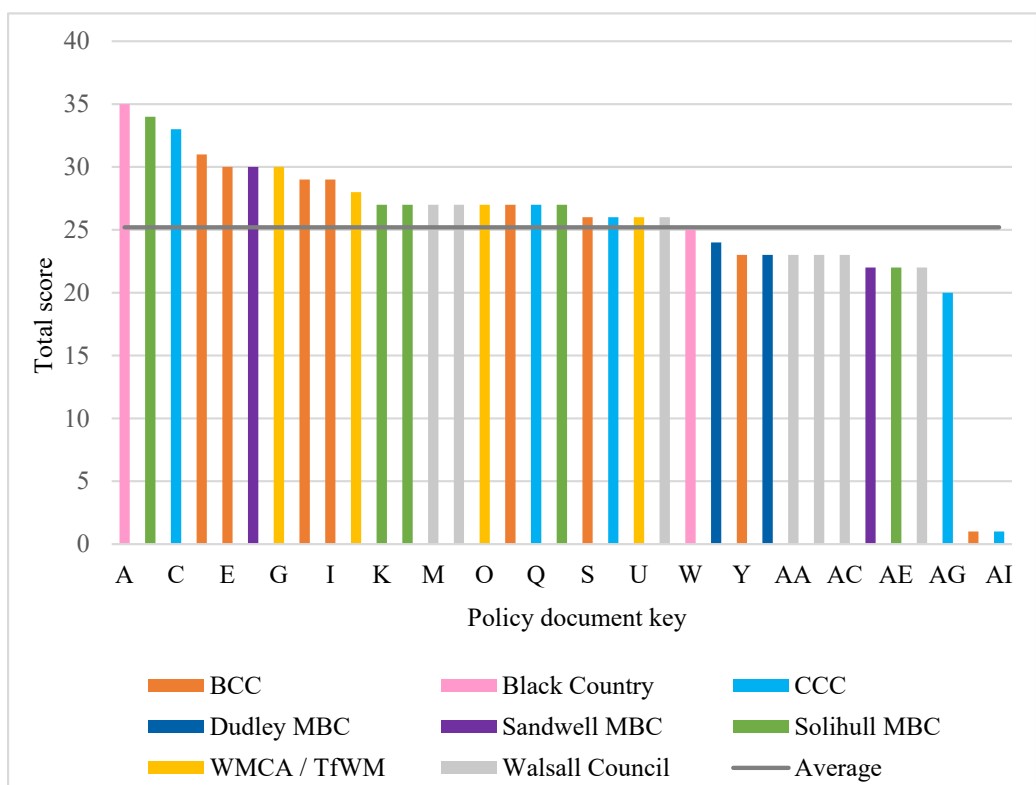

**Figure 2.** Local authority/organisation against total score.

To further investigate the resilience performance of each of the policy documents, they were divided into six categories to determine whether there is any correlation between the type of policy document and the overall average scores, as shown in Figure 3.

'Local Flood Risk Management Strategies' is the highest performing category, with an average score of 32.8. The 'Local Flood Risk Management Strategies' were the only policy type to all score above average whilst the policy documents within the other categories show varying performance.

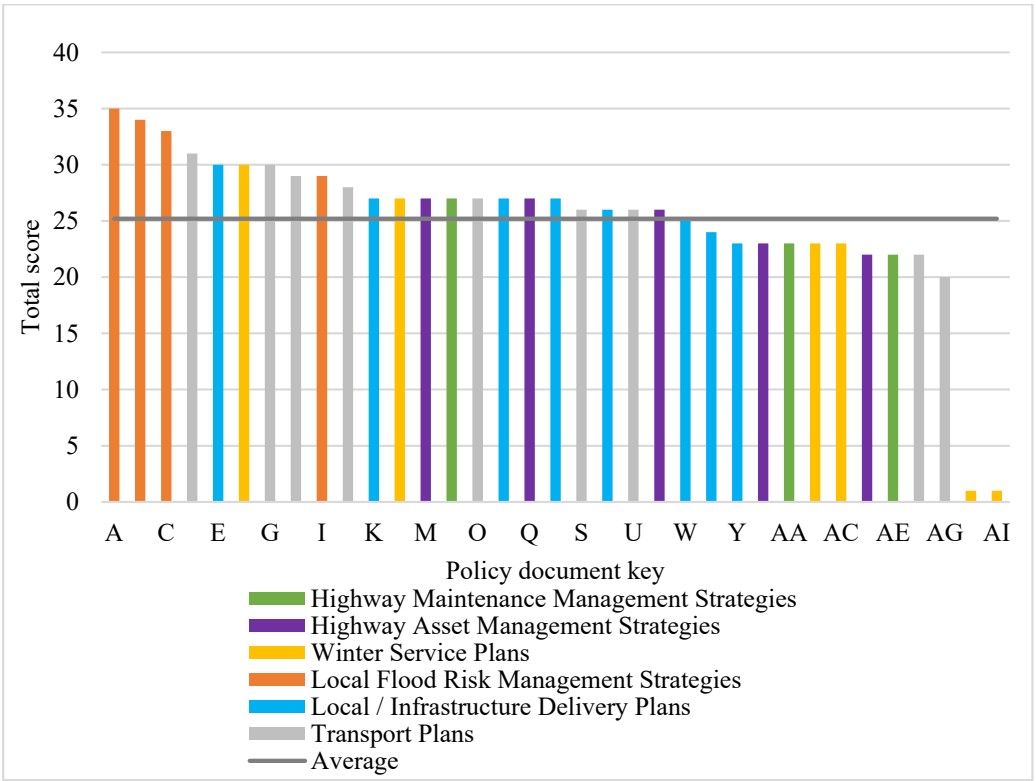

**Figure 3.** Type of policy document against total score.

## 5. Discussion

### 5.1. Resilient Characteristics

As the average performance of each local authority is within a margin of 6.5 points, this discussion focusses on the specific characteristics by which each policy document was evaluated. By highlighting those regional road policy documents that address the characteristics of resilience well, examples of best practice across the region can be identified. The general patterns across the local authorities are presented to contextualise the average performance of these policy documents. By also highlighting those that performed poorly against each characteristic of resilience the evaluation can inform recommendations or local authorities to inform future policy development to support resilient regional road networks.

#### 5.1.1. Roles and Responsibilities

The 'roles and responsibilities' component of resilience gained an average score of 3.3, as shown in Table 2. Several local authorities did not clearly define their roles and responsibilities, instead providing a general statement to convey that it is the responsibility of the Council to carry out the actions that have been identified, rather than specifically stating which member or team within the authority will fulfil each policy. In contrast, the Black Country's Local Strategy for Flood Risk Management 2016 (A) clearly defined the roles and responsibilities of the local authority, as well as listing the roles of its collaborators. The strategy highlighted that there is a range of stakeholders involved in flood risk management, including the Lead Local Flood Authorities, the Environment Agency, and water and sewerage companies. The document also presents an action plan, in which the duties of each partner were outlined (A). This ensures that there is transparency and accountability between all partners.

#### 5.1.2. Regularly Reviewed and Updated

Several policy documents scored 1 out of 5 in the 'regularly reviewed and updated' element of resilience because they simply failed to indicate whether the strategy would

be reviewed in the future. Other documents briefly mentioned that the strategy would be monitored but there was no indication of a timescale to highlight how often the strategy would be reviewed. In contrast, Coventry CC's Highway Infrastructure Asset Management Plan 2019 (Q) emphasised that the strategy should be seen as one that is evolving rather than being fixed and would therefore be reviewed and updated annually.

### 5.1.3. Cost–Benefit Appraisal

None of the policies reviewed showed evidence of a full 'cost–benefit appraisal'. In fact, there were very few cases where the strategies considered any cost–benefit analysis. Birmingham CC's Local Flood Risk Management Strategy for Birmingham 2017 (I) implied that the costs and benefits would be considered by highlighting that the funding level, for each scheme that is paid for by the central government, is dependent on the benefits that the scheme offers. In the case where funding is not granted, local contributions would need to be gathered or the cost of the scheme could be reduced. The strategy also listed other sources of funding that are available, such as the Local Levy (a locally raised source of funding), and then allocated a potential source of funding to each scheme to indicate how it would be delivered (I). Additionally, some of the policy documents were very transparent and stated that there is currently not enough funding to carry out all the schemes within the strategies (H). However, this is precisely the scenario where it is especially important to conduct a cost–benefit appraisal to select the schemes that offer the best value for money to inform the prioritisation of schemes to be funded. It is likely that local authorities do conduct these assessments but may not release this information to the public citing confidentiality reasons. However, a high-level summary of any assessments undertaken should be made publicly available such that stakeholders can be held accountable, and the public can remain informed, even if the technical or sensitive details are removed [18].

Furthermore, Walsall Council's Highway Asset Management Plan 2015–2021 (M) included a table to show how funds have been allocated to the highway network in previous years, which could be useful when predicting future funding needs (M). However, it should be noted that this is a limited approach because the availability of funding can vary each year. For example, an extreme weather event could occur that could significantly impact the level of funding available in future years. There may also be changes in political priorities (B), for instance, COVID-19 has demonstrated that it is sometimes necessary to divert funds to address emerging challenges, as well as affecting the level of funding available from central government. To tackle this issue, the strategy encourages the use of funding to achieve long-term objectives rather than short-term ones (M). The policy documents that performed poorly in the 'cost–benefit appraisal' aspect of resilience failed to provide any indication of the cost to implement the strategy. For instance, Birmingham CC's Birmingham Transport Plan 2020 (S) stated that the level of inward investment is greater than ever, but then did not identify any sources of funding that would be used to deliver the schemes (S). It could be argued that some of the schemes are at the conceptual stage, so it is possible that the costs have not yet been considered in detail.

### 5.1.4. Accessibility

In terms of the 'accessibility' aspect of resilience, most of the strategies performed exceptionally well. The policy documents were overall easy to access on the local authorities' websites and could be understood by a wide range of stakeholders, including members of the public. However, Birmingham CC's and Coventry CC's Winter Service Plan (AH and AI, respectively) scored 1 out of 5 in this category because they did not appear to be accessible on the authorities' websites or on Google Search. Although these two policy documents were not identified in the systematic review, it is a requirement that each highway authority produces a plan for response to winter weather (F), hence they should be available to the public. For this reason, the documents were still included in the evaluation. Other strategies that did not perform as well in this category, such as Solihull MBC's Highway Maintenance Plan 2019 (AE), could not be found on the local authorities' websites. However, they did

appear on Google Search, so they are only accessible to stakeholders who are aware of the specific policy documents that they are searching for.

### 5.1.5. Communication with Road Users

'Communication with road users' was the second lowest scoring aspect of resilience evaluated, since a third of the policy documents had no indication of communicating weather conditions to road users. Some of the strategies mentioned that it is important to share information to the public but did not provide details regarding the means of communication. Other strategies addressed this aspect briefly by outlining that they would inform road users of disruptions but did not specifically reference severe weather conditions. On the contrary, a few of the strategies, such as Walsall Council's Winter Service Operational Plan 2015-18 (AB), expressed that communication regarding current and predicted road conditions is a priority to the local authority, to ensure that timely and accurate information is passed on to the public. This will assist road users with their journey planning decisions, by enabling them to select an alternate mode, route, or time of transport for their journey, and may reduce congestion on the road network. The strategy also provided details of the communication tools it would use to inform road users of adverse weather conditions. These include social media, the Council's website, and variable message signs (AB).

### 5.1.6. Collaboration

The evaluation of the strategies showed that most strategies performed well in the 'collaboration' aspect of resilience, with an average score of 4.7 out of 5, as seen in Table 2. In many of the documents, the local authorities specified that they would need the support of their partners to deliver their vision for transport. The strategies that were awarded a score of 5, such as the Birmingham Mobility Action Plan White Paper 2014 by Birmingham CC (H), included an extensive list of collaborators, which were commonly other local and regional authorities, national bodies (such as National Highways and the government), public transport operators and the Environment Agency (H). The Winter Service Plans and Local Flood Risk Management Strategies also indicated that local authorities are working with the Met Office and MetDesk, who provide weather forecasts that are specific to road conditions (F). This is particularly important because highway authorities must be aware of weather conditions as soon as possible, so that they can respond and restore services in a timely manner. In addition, a partnership with the Met Office and MetDesk enables local authorities to relay weather information to road users promptly, which is another important aspect of achieving resilience. In the very few cases where the strategies performed poorly in this category, the local authorities did not clearly identify their collaborators. However, it can be observed that local authorities have acknowledged the significance of building partnerships and recognise that improving the resilience of local road networks is a shared goal that is much easier to achieve by working collaboratively.

### 5.1.7. Flexibility

The 'flexibility' component of resilience produced a range of results, as shown in the Supplementary Materials (Table S2). Many of the strategies focused on modal flexibility and changing travel behaviours by encouraging the public to use more sustainable means of transport, including cycling and walking. For example, WMCA's Movement for Growth—2026 Delivery Plan for Transport (J) specified that one of its objectives is to see a shift in travel that is in line with large European cities, where car use is significantly lower in comparison to the West Midlands (J). There are also several plans in place to enhance public transport facilities so that they become a more attractive alternative to private transport. This will reduce the number of cars on the road, help to ease congestion, and increase the capacity of the network, which relates to the 'redundancy' characteristic of resilience, as defined in Section 2.1. However, Markolf et al. (2019) draws attention to the fact that during extreme weather conditions, there tends to be a change in travel behaviour from active modes (such

as walking or cycling) to alternate means of transport, such as public transport or cars. This could put additional stress on the road network [30]. It is therefore vital that local authorities communicate weather conditions to the public, as this may result in road users re-routing, retiming, or cancelling their journeys, which may help to reduce congestion.

Moreover, several strategies demonstrated that they offer flexibility in terms of funding. For instance, Sandwell MBC's Winter Service Plan 2019-20 (F) presented a table of fixed costs for winter maintenance that will be incurred regardless of the severity of the winter season, and a table of variable costs that will vary depending on the severity of the season (F). The strategy shows that the local authority has acknowledged that extreme weather events cannot always be predicted and has accounted for this when setting their budget. Similarly, Solihull MBC's Local Flood Risk Management Strategy 2015 (B) outlined factors that may affect the availability of funding, suggesting that local authorities must be able to adapt to unforeseeable changes, such as funding cuts (B). Some of the strategies, such as the Winter Service Plans, have established a priority network to account for the fact that resources are limited (F). Furthermore, very few strategies consider the future of transport and how the road network may be impacted by the introduction of autonomous vehicles. Birmingham CC's Birmingham Transport Plan 2020 (S) does briefly mention that autonomous vehicles will offer significant improvements in efficiency, but there is no further discussion. Several studies do suggest that autonomous vehicles will increase road capacity and could certainly play a part in improving the resilience of road networks [63]. However, they may not be commercially available until 2030 [64], therefore it is possible that policy documents will begin to address the impacts of autonomous vehicles to a greater extent in the future.

### 5.1.8. Interdependencies

'Interdependencies' is the lowest scoring aspect of resilience, with an average score of 1.4 out of 5. Most policy documents did not make any reference to interdependencies with other sectors. A few policy documents, such as Birmingham CC's Birmingham Mobility Action Plan White Paper 2014 (H), mentioned that the local authority will work in partnership with utility companies (H) but did not indicate how the two sectors may rely on each other. In contrast, the Black Country's Local Strategy for Flood Risk Management 2016 (A) performed particularly well because the strategy identified that during heavy rainfall events, there is a possibility that sewer systems may overflow and result in sewer flooding. Severn Trent Water (STW) is the water and sewerage company that is responsible for maintaining the public sewers for the Black Country, and the strategy indicates that the local authorities will collaborate with STW to limit the rainfall that enters the sewers (A). Other strategies, including TfWM's Congestion Management Plan 2018 (U), suggested that there would be an increasing use of technology to monitor the road network and any disruptions it may face (U), but did not consider the implications that this may have in the case of a power outage, for example. Traffic signals and variable message signs also depend on the resilience of power systems, and it could be dangerous if they were to stop functioning. In addition, as electric vehicles become increasingly prevalent, interdependencies with the energy sector must be addressed to ensure that the vehicles can operate reliably [30].

### 5.2. Resilient Regional Highways Policy

It is evident from Figure 2 that Local Flood Risk Management Strategies were the highest performing type of policy document, with the Black Country's Local Strategy for Flood Risk Management 2016 (A) achieving the highest overall score out of all the strategies that were evaluated. The strategies all very clearly defined the duties of the local authority and their partners, were easy to access on the local authorities' websites, and could be understood by a range of stakeholders. The strategies stated exactly how often they would be reviewed, and Coventry CC even indicated triggers that may result in the document being reviewed earlier than planned. Although there was no evidence of a cost–benefit appraisal in any of the strategies, several documents implied that the costs and benefits of the schemes would be considered to determine how funding will be allocated. The

strategies did not perform as well in terms of flexibility but did recognise the importance of sharing weather information with road users through a range of communication tools. All the strategies showed convincing evidence of collaboration with other local authorities and organisations in the West Midlands, and some even highlighted interdependencies with utility companies.

It is interesting to note that the nature of the policy document influenced how well the strategy performed in certain aspects of resilience. For example, some of the documents were more general strategies for the region, such as Solihull MBC's Local Plan: Shaping a Sustainable Future 2013 and Coventry CC's Local Plan 2017 (K and T, respectively). These documents outlined policies for a wide variety of topics including housing and sustainability, as well as the management of road networks. The results indicate that these policies performed better in the 'collaboration', 'accessibility' and 'regularly reviewed and updated' aspects of resilience, however roles and responsibilities were vaguely identified because there are a considerable number of parties involved in delivering the policies. On the other hand, the Winter Service Plans were specific to one component of highway network management. Hence, these strategies tended to define roles and responsibilities very clearly. Furthermore, although there is not a clear correlation between the total score and the local authority that produced the document, as illustrated in Figure 3, it is important that all local authorities share best practice so that they can work towards a shared goal of improving the resilience of the road network.

Recommendations for Transport Policymakers

Reflecting on the findings of the case study there are lessons for many transport stakeholders and policy makers, not only those involved with highways, as well as for local authorities beyond the West Midlands. The key recommendations are listed below:

- Recommendation 1—Local authorities should be encouraged to produce a dedicated road network resilience strategy.
- Recommendation 2—To address budget constraints, local authorities should conduct cost–benefit appraisals of resilience strategies to determine which of their schemes offers the best value for money.
- Recommendation 3—Local authorities should ensure that their strategies include a complete list of their partners and clearly define their roles and responsibilities to ensure that there is accountability across all parties.
- Recommendation 4—Local authorities must consider interdependencies between sectors, to ensure that a disruption to one network does not impact upon another.
- Recommendation 5—It is vital that local authorities take a long-term approach to achieving resilience due to the unpredictable nature of weather events. Local authorities must learn from past disruptions and should strengthen their strategies to prepare for weather conditions that they may have never experienced previously.

*5.3. Approach, Limitations and Future Work*

5.3.1. Resilience Characteristics and Evaluation Criteria

This study adapted existing approaches [24,60] to identify key criteria for resilient transport systems and regional highway network management strategies for evaluation. The evaluation criteria were identified from peer reviewed academic literature and supplemented by other literature, which is accepted practice. Another approach to consider for robust evaluation criteria is using the International Standard Organisation's (ISO) documentation [65]. In the absence of an ISO standard which addresses resilience to climate change, evaluation criteria from ISO 14,090:2019—Adapting to climate change [66] could be used to provide a range of evaluation criteria [67]. It is worth noting that these bear some resemblance to the criteria identified in this paper. The general adaptation principles of ISO 14,090 are:

1. Change-oriented perspective
2. Flexibility
3. Mainstreaming and embedding
4. Robustness
5. Subsidiarity
6. Sustainability
7. Synergy between adaptation and mitigation of climate change
8. Systems thinking
9. Transparency
10. Accountability

### 5.3.2. Resilient Policy

The regional highways policies were identified through a systematic Google Search as there is no comparable online database to support the systematic review of policy documentation. Whilst every effort was made to get copies of the policies identified some of the policies were unavailable to the author, however they were still included in the evaluation. Without access these policies could not be fully evaluated across all the resilience criteria, however, they could be evaluated for their accessibility, for which they scored exceptionally low marks. It should be noted that there may be additional policies not made available to the public, as such these have not been included in the review.

The approach developed has been applied in a case study of West Midlands Highways strategy and policy documentation, but it is sufficiently flexible that it could be applied to any transport policy, anywhere in the world and at any scale (regional, national, international, etc.). However, the resilience criteria evaluated may not be sufficiently detailed to reflect the current maturity of some fields or locations. Consequently, further work may be needed to adapt this approach to address specific users' needs.

### 5.3.3. Resilience of Regional Transport

The results of the analysis undertaken here only reflect the maturity of reginal policies in the West Midlands in the UK. As the largest conurbation outside of London, the West Midlands provided a useful test-bed to explore and develop this type of methodology. However, these outputs could be more beneficial if they accounted for a wider range of organisations and strategies. Ultimately extending this approach for wider geographical inclusion across the UK may provide an effective tool to instigate collaboration between stakeholders and implementation of best practices. One major challenge to this may be the differences between policy structure across different regions and local authorities where the members of the WMCA had many of the same documents with the same purpose. However, those documents authored in collaboration (WMCA, TfWM or the Black Country) tended to score higher overall, indicating this kind of collaboration can lead to greater performance through consistency of messaging whilst still addressing transport resilience challenges at a regional scale.

### 5.3.4. Multi-Criteria Analysis

Whilst multi-criteria analysis such as that presented in this evaluation is often weighted to consider the proportional contribution of each of the variables this was not done during this study. The evaluation undertaken here would not benefit from the addition of weightings as there is a limited number of criteria [61]. Further analysis, with increased complexity may find it suitable to include a criteria weighting and sensitivity analysis.

### 6. Conclusions and Forward Look

Improving the resilience of local road networks to climate change is a complex process for several reasons identified here: the impacts of climate change are not always predictable; the funding that is available is often insufficient; the road network is usually designed to operate in climate conditions that have now been exceeded; transport demand is increasing

and will need to be accommodated for in the future; and technology is evolving rapidly resulting in growing interdependencies between sectors. However, transport policy provides the framework within which these issues can be addressed, but it can only be effective if policy is evaluated against the needs of the systems to which they relate.

The methodology presented provides a novel, scalable and practicable approach to evaluating the efficacy of policy and strategic planning documents in addressing climate change resilience. Whilst this approach has been developed in the context of regional highways strategies the method is sufficiently agile that it could be adapted to apply in other settings or at different geographical scales. This flexibility is afforded at the expense of a comprehensive weighted multicriteria analysis, however this can be addressed with future work to extend the method for increased detail and a wider geographic purview. This will become increasingly necessary as climate change resilience policy develops over the coming years.

Whilst the evaluation criteria of the multi-criteria analysis were applied to a case study in the West Midlands in the UK the results still outline the key aspects that should be accounted for to address the resilience of transport to the impacts of climate change. These outputs are therefore informative for a wide range of policymakers and transport stakeholders and form the basis of future robust reviews of the effective consideration of resilience to climate change within transport policy and beyond. In addition, the method developed may inform further, in-depth, academic study of infrastructure policy provision through a climate change resilient lens.

**Supplementary Materials:** The following supporting information can be downloaded at: https://www.mdpi.com/article/10.3390/infrastructures7110146/s1, Reference List S1: List of policy documents gathered through systematic review; Table S1: Policy Document Key; Table S2: Policy Evaluation Results Table.

**Author Contributions:** Conceptualization, R.S.F. and S.B.; methodology, R.S.F., S.B. and A.D.Q.; formal analysis, S.B.; writing—original draft preparation, S.B.; writing—review and editing, R.S.F. and E.J.S.F.; supervision, R.S.F. and A.D.Q.; project administration, R.S.F. All authors have read and agreed to the published version of the manuscript.

**Funding:** Rachel S. Fisher's time on this research was in part funded by NERC grant NE/X001938/1. Emma J. S. Ferranti acknowledges EPSRC Fellowship EP/R007365/1.

**Data Availability Statement:** The data presented in this study are available in www.mdpi.com/xxx/s1.

**Conflicts of Interest:** The authors declare no conflict of interest.

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
