# Peer review of "Evaluation of Climate Change Resilience of Urban Road Network Strategies"

_infrastructures, doi:10.3390/infrastructures7110146_

Round 1

Reviewer 1 Report

The manuscript has a number of grammatical errors and typos.

The formatting of the text should be checked in accordance with the journal template.

The novelty of the research should be better specified in the introductory part

Limitations and future steps of investigation should be better emphasized in the concluding part 

in Table 1 criteria are defined for which it is advisable to include reference literature ,by inserting a column in the table for example .

More comments should be included to accompany Figures 2 and 3 

All acronyms should be written in expanded form when they first appear in the text 

Specify whether the assessment conducted is replicable in other contexts. 

It is probably appropriate to include something relevant to the proposed methodology among the key words

Lines 90 to 109 could be better interpreted if included in a flow chart describing the steps of analysis conducted.

Author Response

Following the reviewers feedback the paper has undergone an extensive edit. It is possible that I've introduced new issues with regards to typos and formatting however I'm hoping that you will find the content vastly improved. Please see the attached document for the responses to both reviewers comments. 

Reviewer 2 Report

Reviewer comments:

Main comments and major concerns:

The authors have conducted a literature review and delineated five recommendations that road authorities, isolated to the West Midlands in England, may implement to increase the resilience of their road networks. The research is not a systematic review as it does not abide with the methodological approach of a systematic review, but rather conforms to a narrative review. The research has a faint golden thread that could be greatly improved. The relevance of the research to the wider academic world is uncertain, as it is never discussed. The research findings are extremely limited as currently presented. Clearly these recommendations could be applicable to all road authorities in a generalised manner, but this has not been engaged with. The authors have not provided any real baseline for comparison, such as National Highways, Transportation Infrastructure Ireland, Federal Highway Administration, or similar global road authority leaders, and it is not certain how the West Midlands compares to other locals (i.e. Kent Council). This is to be expected given the isolation of the research and system boundaries, which is fine but this limitation should then be stated. A small evaluation could be very beneficial though. With that being said, the main concern is that of “proactive maintenance”, the leading strategy to ensure resilience against climate change of road networks, and is accompanied by a wide variety of economic, social and environmental benefits. A broader literature review not isolated to the West Midlands would have captured this. If it does not, the authors need to explain why it is not considered a leading recommendation.

Minor and generalised comments:

Line 35 – “impact global pandemic” – grammar. A handful of other grammatical errors are present throughout the paper. I have not commented on each.

Authors keep referring to the pandemic and COVID-19. Everyone is aware of the pandemic. The authors keep making arguments of what it was like prior and fall short of carrying the argument further. Is reference to the pandemic essential, or can 2019 as a date just be referred to rather? If the research is only based on pre-pandemic occurrences and relevant to the pre-2019 world, it is already irrelevant as the world is post-pandemic now.

Line 50 – pretty sure the last few weeks broke the 2019 record.

Line 70 – be careful of making such comparisons and leading statements, that 2% contributes well over 50% of total traffic volumes and most likely over 80% of road freight (i.e. HGVs).

Line 75 – CCRA? Be consistent with acronymising. Check others as well and ensure correctness.

Line 232 – This already happens on most SMP schemes using traveller information systems (i.e. MS4s etc.) as well as use of social media? Describe this better

proactive maintenance is perhaps the best strategy to enhance the resilience of the UK road network.

Methodology: general comment – there are a lot of sentences that repeat the process the authors followed. There is a fine line between explaining something to a layman and treating the reader as if they are uneducated. The latter is obviously not recommended.

Line 353 – “Table 1,…” remove comma

Line 361 – reason for not applying a weight is poor. It suggests the authors did not want to engage with it enough and found an easy way out. Either find a better reason to omit weighting, or add a weighting.

Methodology: Justify why only Google search was used (obviously not a common approach among reviews) - i.e. council documents are probably available on Google only and not Google Scholar for instance...justify it

Table 1: what about 2-4 scores? Not good enough to expect the reader to fill in the gaps. Table 1 format is also different than the general text and not consistent.

Section 4: “The criteria outlined in Section 3.3 are used to evaluate how effectively a range of regional policy documents capture the aspects of resilience that have been determined to be important based on the systematic review in Section 3.1.” – says who? Is this an approved international method of evaluation or one the authors developed themselves? Not clear – look at https://doi.org/10.1016/j.treng.2021.100049 Appendix: Supplementary Information for ways around this. Adapting existing peer-reviewed and published matrices to fit your research is advised.

Line 602 to 606 – good reasoning for a clear study limitation. Apply the same to other aspects of the manuscript.

Recommendation 2: be aware of the implications of full transparency. HS2 and SMP are good examples of what happens when the public knows what the projects cost. Not saying don’t promote it but be aware of the impact transparency can have (e.g. wide scale protest).

Funding: not necessary to state you are a Master’s student. Many post graduate studies are funded so it is a flawed argument anyways.

Author Response

(The authors gave the same response as above.)

Round 2

Reviewer 1 Report

The manuscript has several grammatical errors .In the introductory part, the concept of "Climate Change Resilience of Urban Road Network Strategies " needs to be better explained by including mentions of Agenda 2030 and the drafting of SUMPs

High-resolution images need to be included Some acronyms need to be made explicit again in expanded form 

More details and explanations is needed to accompany Figures 2 and 3 

In the initial part, it is necessary to better point out the research steps conducted and the derivation of the parameters entered in the tables (by including more literature references for example))

Reviewer 2 Report

The authors have now engaged with the essence of the topic, looked at the broader picture and the paper now has sufficient weighting to contribute to the scientific community and expand the discussion on climate resilience of our road networks which is urgently needed. Final comment is to consider removing acknowledgements focused on reviewer inputs. You do not need to thank us in the paper, we review out of our own free time and professional commitment to the academic world, and our papers are equally reviewed by others who do the same. Best of luck with your future research endeavours.

Author Response

I've uploaded the response to reviewer 1's comments for transparency.

Thank you for your supportive comments, I think the article is much stronger now.

Round 3

Reviewer 1 Report

the manuscript still has numerous typos and grammatical errors 

More bibliographical references need to be included to accompany Table 1 and paragraph .

We recommend reading 

1) Ezgeta, D., Čaušević, S., & Mehanović, M. (2022, May). Challenges of digital transport transformation in Europe. In FIRST INTERNATIONAL CONFERENCE ON ADVANCES IN TRAFFIC AND COMMUNICATION TECHNOLOGIES (p. 37).

2) Cruz, C. O., & Sarmento, J. M. (2020). “Mobility as a service” platforms: A critical path towards increasing the sustainability of transportation systems. Sustainability12(16), 6368.

3) Campisi, T., Georgiadis, G., & Basbas, S. (2022). Developing Cities for Citizens: Supporting Gender Equity for Successful and Sustainable Urban Mobility. In International Conference on Computational Science and Its Applications (pp. 410-422). Springer, Cham.

It is necessary to check the numbering of the references and their template

It is necessary to revise the formatting of Table 2 in order to make the parameters analyzed more understandable 

It is recommended to delete the titles in the charts and insert such text as a caption 

Author Response

Thank you for your further review. 
The article has been read by senior members of the writing group. They are happy with the text. If there are specific issues please could you highlight these otherwise we, as a team, don’t believe there to be any further textual changes required. The changes made can be seen in the tracked changes.

Table 1 is from the resilience shift report on public policy this has now been made clearer in the table heading with text alongside the reference. We do not believe that this requires further references as this is presenting a general high level example. Whilst we recognise that the issues of digitalisation, MaaS and gender equity are very important (and I've enjoyed reading this literature, thank you for highlighting it) they are not closely related to the purpose of Table 1 as such these have not been included. To do so at this stage would require significant further work.

The references use the MDPI mendeley referencing template and the font now aligns with the formatting provided in the MDPI paper template. I've also double checked this against recent publications and I believe this is now correct. See tracked changes. I have double checked the numbering of the references and believe they are correct, if there is a specific concern could you please identify this?

Table 1, 2 and 3 have had adjustments to their formatting to make these more easy to read.

Figures 2 and 3 have both had the titles removed from the chart and caption text adjusted. I have also reformatted these to make better use of space. I will reupload the graphics without the titles now.